# The program efficiency of environmental and social non-governmental organizations: A comparative study

Sujie Peng *

Faculty of Business and Management, Beijing Normal University-Hong Kong Baptist University United International College, Zhuhai, Guangdong, China

* sujiepeng@uic.edu.cn

## Abstract

Non-governmental organizations (NGOs) are becoming increasingly significant stakeholders in global governance and business operations. However, measuring their efficiency is a challenging task due to their mission-driven nature. While previous research has proposed financial and non-financial indicators to measure NGO efficiency, none of them has compared the differences between environmental and social NGOs. This study aims to investigate the factors influencing the program efficiency of NGOs in China and compare the differences between environmental and social NGOs. 12 indicators are employed and tested using data collected from the Chinese Research Data Services (CNRDS) platform. The study employs multiple regression analysis to examine the influential factors identified in the dataset. The findings demonstrated different influential factors of program efficiency among environmental and social NGOs. The results of the analysis provide valuable insights for NGO operators, policymakers, and researchers in the field of NGOs.

## Introduction

In recent times, non-governmental organizations (NGOs) have ascended to unprecedented levels of influence. According to estimates provided by the World Bank [1], the cumulative value of global transactions involving NGOs attained a remarkable $2.3 trillion in the year 2016. Nevertheless, these organizations are currently confronted with intensifying contestation for financial resources. This funding is derived from a wide array of sources, encompassing individual donors, foundations, and governmental bodies. This has brought about a sense of urgency for these organizations to be accountable and measure their performance. However, many performance reports and measurement systems tend to focus solely on financial measures, such as donations, expenditures, and operating expenses. The true success of a nonprofit should be measured by its effectiveness and efficiency in meeting the needs of its constituents, with financial considerations playing a secondary role. The act of performance measurement is deemed a pivotal strategy for NGOs to manifest their efficacy, performance, and values, with the aim of preserving public confidence and ensuring a continuous inflow of funding and resources [2]. Furthermore, it is of paramount importance for these entities to employ the

**Editor:** D. Daniel, Gadjah Mada University Faculty of Medicine, Public Health, and Nursing: Universitas Gadjah Mada Fakultas Kedokteran Kesehatan Masyarakat dan Keperawatan, INDONESIA

**Data Availability Statement:** The data employed in this investigation are proprietary to the CNRDS

database (https://www.cnrds.com/Home/Login). Acquisition of the requisite access rights is feasible for any investigator through either institutional affiliation or individual registration and procurement.

**Funding:** This work was supported by the UIC Start-up Research Fund [grant number UICR0700043-23]. The funders had no role in study design, data collection and analysis, decision to publish, or preparation of the manuscript.

**Competing interests:** The author has declared that no competing interests exist.

most appropriate forms of evidence during the processes of development, implementation, and evaluation of their services [3].

Under the context of NGOs, efficiency can be conceptualized as meeting its mission at the lowest cost. Numerous studies, such as [4, 5], have delved into the exploration of measures that constitute NGO efficiency. They've examined several indicators including the fundraising ratio, project ratio, allocative efficiency, and technical efficiency. Yet, due to the service-based nature of NGOs, efficiency measurement is still challenging [6]. In particular, measuring efficiency in the nonprofit sector is challenging due to several reasons, including the non-commercial nature of NGOs' activities, confusing terminologies, and unsatisfactory cost-benefit ratios. The fundraising efficiency ratio, for instance, calculates the ratio between fundraising expenditures and income from donations, while the program efficiency ratio measures the percentage of resources allocated to projects related to NGOs' missions in a period [7]. To our knowledge, in the existing works, studies regarding the assessment of NGO efficiency are lacking.

Moreover, the missions of NGOs are often complex and multifaceted, as they are shaped by the perspectives and priorities of various stakeholders involved in the organization [8]. NGOs have been categorized based on their mission focus, specifically into environmental NGOs (ENGOs), which primarily tackle environmental concerns, and social NGOs, which predominantly address societal issues. In particular, there has been a significant amount of research conducted on the influence of ENGOs as a form of informal environmental regulation on environmental quality. The literature has examined various examples, such as the role of ENGOs in environmental governance in China [9], and enhancing the ecological environment of the Baltic Sea and Adriatic-Ionian Sea regions [10]. Furthermore, NGOs play critical roles in dealing with social problems, such as poverty reduction [11] and labor rights [12]. In essence, it can launch initiatives, such as those implemented by the Rainforest Alliance [13], that enhance the social sustainability of communities by educating and integrating impoverished producers into corporate supply chains. The outcomes of these projects have led to diminished occurrences of child labor, improved financial gains for indigent producers, and increased access to employment opportunities for women. To the best of our knowledge, none of existing studies compared how the influential factors of efficiency between environmental and social NGOs.

Ragin [14] posits that the objective of comparative research is to elucidate and interpret macro-social variation. This form of research aims to pinpoint and explicate the economic, political, sociocultural, and historical parallels and/or disparities in a given phenomenon across diverse societies. As underscored by von Schnurbein et al. [15], the utilization of comparative research methodologies in the examination of NGOs has been a fundamental approach in extending our understanding of these institutions. Current literature has conducted comparative analyses at both the national and sectoral levels. For instance, Bercea et al. [16] suggest that comparative analysis can be conducted at the organizational level, allowing for a comparison between for-profit and nonprofit organizations. Similarly, Saqib et al. [17] indicate that this approach can also be applied at a national level. Comparative analysis offers benefits such as enhancing concepts, revealing specific and general forces behind a phenomenon, and creating guidelines to improve practices [18]. Thus, it's a valuable tool for deepening our understanding of NGOs. The capacity for comparative analysis has stimulated inquiries into the catalysts behind the global expansion of NGOs, the factors enabling these organizations to adapt to diverse contexts, and the subsequent implications for our societies.

However, there's a scarcity of studies discussing this phenomenon at the organizational level in non-western settings. On one hand, environmental nonprofits are non-government, non-commercial groups focused on issues like sustainability, wildlife protection, biodiversity

management, environmental research, policy advocacy, and conservation efforts [19]. Conversely, social NGOs or socially oriented NGOs, as indicated by Kataeva et al. [20], concentrate on addressing societal challenges, including child labor, working conditions, and health and safety issues, as noted by Yawar and Seuring [21]. To my understanding, again, there is still a limited comprehension of the factors influencing the efficiencies of both types of NGOs, as well as the practical and social implications of these influences. Hence, a comparative analysis of the efficiency between environmental and social NGOs in China is necessary. Therefore, our study aims to address the following question:

*RQ 1*: *What are the influential factors of NGO efficiency in China*?

*RQ 2*: *How do the influential factors of program efficiency are different between environmental and social NGOs*?

The rest of this article is organized as follows: the next section reviews related literature; this is followed by a section outlining the methodology used in this study; subsequent sections present empirical results and discussion; the conclusions are presented in the final section.

## Literature review

### The nonprofit sector in China

China's nonprofit sector has grown significantly over the past forty years due to a shift from exclusivism to corporatism, now encompassing areas like environmental protection, poverty reduction, education, and healthcare. According to China's Ministry of Civil Affairs [22], by 2016, registered nonprofits in mainland China exceeded 700,000, a 300% increase since 2000. These nonprofits employed 7.64 million full-time staff and received 78 billion RMB in donations. Furthermore, the count of unregistered nonprofits in China has also significantly risen in recent years. Estimates suggest there are over 1 million such organizations [23]. Regrettably, no precise data exists to monitor these entities.

In China, one category of NPOs is NGOs. Officially, in China, the term 'NGO' often called 'social organization,' includes all non-profit entities outside the state system [24]. These are independent nonprofits established to address social needs, like social services or civic organizations [25]. These NGOs are essentially non-profit organizations not managed by the government, although many are government-organized NGOs (GONGOs). Typically, these organizations are smaller and operate independently from the government. Unlike GONGOs, which depend on relationships with specific government agencies, NGOs compete for private funding to survive. This competition hinges on professional ability and accountability, not on the influence of government sponsors [26]. Official NGO registration in China requires finding a government department willing to sponsor the application. However, as NGOs often operate outside state institutions and could pose risks to sponsoring departments, few incentives exist for this task [27]. Consequently, many NGOs look for alternate registration methods to gain legal status. For instance, they might register as a for-profit entity with the State Administration for Industry and Commerce but continue their NGO operations [27]. In conclusion, the context in which NGOs operate in China differs from the Western context, yet these organizations are still striving to improve their efficiency to achieve their missions.

### NGO performance measurement

Although the nature of NGOs and firms is different, the application of PM borrowed from the private sector can be extended to the context of NGOs [28]. NGOs are now expected to provide evidence of their program performance, impact, and outcomes, in addition to financial

performance, to demonstrate their effectiveness in achieving their social mission. The need to measure the performance of NGOs has been driven by various factors, such as meeting the expectations of governmental agencies, donors, and other stakeholders [29]. Donors, in particular, tend to exert significant pressure on NGOs to demonstrate their performance. The measurement of performance is crucial for NGOs as it serves two main purposes: to provide evidence of the organization's value and to benchmark performance for program and service improvement [29]. Simply put, by focusing on performance improvement and assurance, NGOs can prove their worth to donors and stakeholders, and ensure their long-term sustainability.

However, measuring the performance of NGOs can be challenging due to their social mission nature, as indicated by several scholars [30, 31]. Their interventions' activities and outcomes are intertwined, creating a complex relationship that makes it difficult to measure their performance. While funding or fundraising efficiency is a classic technique to measure NGOs' performance, it does not provide a comprehensive evaluation of their financial performance. In addition to their ability to access funds, NGOs' financial activities and their display of financial transparency, key aspects of their financial performance, should also be assessed [32, 33].

Moreover, Polonsky and Grau [34] categorized these approaches into four groups: operating efficiency, organizational objective achievement, return on investment, and social outcomes. Efficiency in the nonprofit sector can be assessed through operating efficiency, which involves the allocation of organizational funds, including the percentage spent on social objectives [5]. Some countries mandate performance standards for NGOs, such as limiting the percentage spent on fundraising. Goal-based assessments focus on whether NPOs are achieving their social goals. NGOs can also use methods like social accounting [35] to put a dollar value on their social activities. Additionally, social outcomes are an important aspect of NGOs, where the focus is on improving social activities. Nonetheless, given that NGOs encompass a spectrum of social objectives with subjective metrics, drawing comparisons across differing issues presents a considerable challenge [34].

Efficiency measurement is a crucial aspect of organizational management as it indicates how well an organization can convert inputs into outputs. Efficiency in the context of NGOs refers to their ability to use their limited resources to achieve their goals and deliver their services effectively [36]. To clarify, efficiency in the nonprofit sector is not solely about cost reduction. Rather, it prioritizes accomplishing the organization's mission within the constraints of available resources. This necessitates that NGOs strategically utilize their resources to fulfill their objectives, instead of merely focusing on cost-cutting. Thus, in seeking to evaluate the degree to which NGOs direct their financial resources toward the realization of their declared aims and objectives, we opted to use the proportion of financial resources expended on mission-related projects as our metric of assessment.

## NGO professionalism

Numerous studies have established a positive correlation between NGO professionalism and efficiency. Selden and Sowa [37] posit that professionalizing human resources, coupled with cultivating a skilled, engaged, and paid workforce, not only elevates personnel satisfaction and service orientation but also reduces turnover and aligns with organizational objectives, thereby driving productivity. Similarly, drawing from Kreutzer and Jäger [38], heightened professionalization, characterized by a focus on efficiency, fundraising, formalization, control, and reporting, is expected to enhance revenue generation and cost reduction. Consequently, this could decrease the cost per output or asset-per-beneficiary ratio, and improve the labor productivity indicator, defined as revenue per unit of human resources. Also, as per Ni et al. [39],

professionalization generally boosts fundraising efficiencies in private foundations, particularly when raising unrestricted funds, a trend not observed in public foundations.

Conversely, a different set of studies has discovered a negative correlation between NGO professionalism and efficiency. With an escalation in professionalization, there is a greater propensity for the managerial persona to supersede the volunteer persona. While the volunteer persona is anchored in an altruistic paradigm, emphasizing democratic engagement and the accomplishment of the organization's mission, the managerial persona adheres to a paradigm of formalization, specialization, and efficiency [38]. Therefore, two potential adverse outcomes of professionalization within NPOs include goal displacement, which is linked to the self-interest logic prevalent in for-profit entities [40], and the surfacing of conflicts due to formalization's significant negative influence on volunteer motivation [38]. Different from the abovementioned studies, Sanzo-Perez et al. [41] substantiate a 'U-shaped' association between professionalization and an NPO's capacity to cater to a larger beneficiary base with fewer resources. Furthermore, professionalization exerts a positive impact on revenue generation, and collaborations with commercial entities augment the non-profit's resource-to-beneficiary ratio.

## NGO political connections

As previously noted, the establishment of political ties between Chinese NGOs and the State engenders a multitude of organizational advantages. These advantages encompass enhanced access to information, regulatory leniency, and bolstered economic performance [42]. Chinese NGOs exhibit strategic acumen in managing their relationships with the State, employing a diverse array of approaches to cultivate and sustain political guanxi. Despite the legal requirement for NGOs to possess a sponsoring government agency, they frequently foster relationships that extend beyond the stipulated mandate [43]. A prevalent strategy employed by NGOs to instigate relationships with the State involves inviting former high-ranking government officials to join their ranks, serving in capacities such as board members, presidents, honorary presidents, or part-time staff members [44]. Lu [45] characterizes this strategy as particularly potent, given that the management of NGOs in China is predicated on the "rule of men" (renzhi), as opposed to the "rule of law" (fazhi). NGOs leverage their political connections to broaden the scope of activities they can undertake, surpassing the constraints imposed by existing legal regulations [43].

## Hypothesis development

### NGO professionalism and program efficiency

Nonprofits are staffed by both paid employees and volunteers, leading to questions about organizational identity and culture [41]. The degree of professionalization within nonprofits varies, but many organizations have increased staffing and resources to keep up with the competitive environment [46]. From the knowledge-based perspective, knowledge is the only resource that provides a sustainable competitive advantage [47]. Professionalization examines the extent to which an NGO has a division of labor and the specification of responsibilities and positions (Hwang and Powell, 2009). In summary, the concept of professionalization within organizations can be interpreted as the acquisition and refinement of skills among the workforce.

Under the context of NGOs, professionalization can be reflected by the increasing recruitment of full-time employees [48]. For example, the possibility of securing government funding can be improved when nonprofits recruit fewer volunteers [49]. However, in the existing works, the impact of the professionalization of human resources on NGO efficiency is mixed.

On one hand, Striebing [50] asserts that the principal driving force for voluntary transparency in organizations is professional management, such as the hiring of full-time personnel, which leads to higher overall efficiency [37]. Yet, on the other hand, the pursuit of professionalization may cause the ignorance of volunteers within NGOs [38]. In other words, the unbalanced relationship between paid staff and volunteers may lead to ineffective human resource management within the nonprofits. Differing from these works, nonlinear relationships can be also identified. For instance, Sanzo-Perez et al. [41] pointed out a nonlinear relationship between professionalization (e.g., professionalized human resources) and a nonprofit organization's ability to serve more beneficiaries with fewer assets, showing a U-shaped curve.

In a similar vein, Suárez [49] also pointed out that professionalized organizational structure design can improve NGOs' possibility of securing funding provided by governmental agencies. It provides organizations with knowledge for successful project execution [51].

Once again, more professionalized NGOs are more likely to adopt an effective internal governance mechanism (e.g. efficient organizational structure) and mobilize external resources (e.g. qualified staff with extensive social connections). In essence, professionalization not only enhances NGOs' governance but also fosters balanced stakeholder relationships and minimizes conflicts of interest, facilitating collaborative projects for improved program efficiency. Therefore, the following hypothesis is developed:

*H1*: *NGO professionalism is positively associated with efficiency;*

*H1a*: *ENGOs' professionalism is positively associated with their program efficiency;*

*H1b*: *Social NGOs' professionalism is positively associated with their program efficiency;*

## Political connection and program efficiency

In many studies, organizational performance relies on not only organizations' internal resources but also their social networks. In the context of business studies, many studies have found a positive association between the quality of social capital and firm performance. Specifically, the network can comprise political ties, such as informal connections with government officials and organizations [52], or business ties, like informal connections with other business organizations [53]. Not coincidentally, a positive linkage can be found between having connections with former politicians and firm performance [54]. However, according to Wong and Hooy [54], firm performance positively correlates only with stable political connections, such as those with government-linked companies and boards of directors, but not with less stable connections like those with businesspeople or family members. In other words, firms are more likely to operate effectively by cultivating boards whose members have political experience and by recruiting directors who also have strong political experiences [55]. More recently, Lee [56] investigated the operations of Chinese family businesses and found that the connections between families and other firms are more likely to enhance firms' networks and performances. However, political connections negatively impact the performance of family businesses [56]. In particular, especially in the Chinese context, scholars have identified the impacts of political connections on organizational performance. The predominant understanding is that, in the not-for-profit sector, political ties can contribute to program efficiency.

In our context, political alignment makes positive impacts on NGOs' efficiency. Indeed, as Ni and Zhan [57] established, nonprofits with strong political ties have an advantage in securing government subsidies, considering the government's extensive influence over crucial decisions such as resource allocation, information access, and legal protection. Moreover, as previously mentioned, increased financial resources enable NGOs to hire professionals and

external consultants to enhance their internal governance, thereby improving the quality of their services. Concurrently, political connections offer NGOs a more stable environment and resources for their activities. Based on these premises, the following hypothesis is proposed:

*H2*: *With stronger political connections, NGOs are more efficient;*

*H2a*: *With stronger political connections, ENGOs are more efficient;*

*H2b*: *With stronger political connections, social NGOs are more efficient;*

## Methodology

This study aimed to address the aforementioned issue through the use of datasets obtained from a research database. The following section provides a rationale for the methodology employed in this study.

### Variables

In this section, I explicate the reasoning for the selection of variables in my study. I employed a total of 12 variables, grouped into four categories: (1) NGO professionalism, (2) political connections, (3) control variables, and (4) NGO efficiency. A detailed description and categorization of these variables can be found in Table 1.

As shown in the three panels of Table 1, this study utilized 12 variables. Generally, all of the variables were chosen in accordance with guidelines set in previous studies. For instance, based on Coupet and Berrett [58]and Harris et al. [59], *Efficiency*, represented by the percentage of funds that NGOs allocate for mission-related purposes, serves as an independent variable to gauge NGO efficiency.

**Table 1. Variables and definitions.**

| Name | Variable | Attribute | Definition |
|---|---|---|---|
| **Panel A: NGO Professionalism** | | | |
| Feedback | The score obtained from performance feedback | Categorical | NGO's rating in the national evaluation |
| Staff | Number of staff | Continuous | Total number of full-time and part-time staff |
| Volunteer | Number of volunteers | Continuous | Total number of volunteers |
| Income | Annual income | Continuous | NGO's total amount of annual income from private sources |
| **Panel B: Political Connection** | | | |
| Registration | Registration level | Categorical | Whether the NGO is affiliated with a low- (e.g., municipal government agency), middle- (e.g., provincial government agency) or high-level governmental agency (e.g., national government agency) |
| Current_Gov | Number of senior managers who are working in the provincial official institutions | Continuous | Total number of NGO managers who are currently working for the provincial governmental organization |
| Previous_Gov | Number of senior managers who have past work experience in the national official institutions | Continuous | Total number of NGO managers who were working for national governmental organizations in the past |
| Gov_Fund | The total amount of governmental funds | Continuous | NGO's total amount of subsidies from governmental organizations |
| **Panel C: Control Variables** | | | |
| Yr | Number of years since the organization was founded | Continuous | The duration in years since the establishment of the NGO |
| Scope | Scope of operation | Binary | Regional scope for NGO's operation (i.e. whether the NGO is operating nationally or locally) |
| Start_Fund | Startup capital | Continuous | The aggregate financial resources available to the NGO at the inception date |
| **Panel D: NGO Efficiency** | | | |
| Efficiency | The proportion of funds expended on project implementation. | Continuous | The proportion of funds allocated towards the execution of mission-related projects by NGOs |

The dependent variables, as previously stated, encompass NGO professionalism and political affiliations. The measurement of NGO political connection aims to capture the extent to which NGOs have developed strong networks with government agencies. In the context of this study, based on Johnson and Ni [60], it is measured by four indicators: "*Registration*" (NGO registration level), "*Previous_Gov*" (number of staff who have worked in provincial governmental organizations), "*Current_Gov*" (number of staff who are currently in national governmental agencies), and *Gov_Fund* (NGOs' total funds from the government). Finally, the control variables were selected based on Wang [61]. In this sense, the following variables were selected in this study: *Yr* (number of years since NGOs started operations), *Scope* (whether NGO is operating nationwide or locally), and *Start_Fund* (NGO startup capital).

## Data collection and sampling

The present study utilized data obtained from the Chinese Research Data Services Platform (CNRDS), a reputable database that offers high-quality quantitative data for research purposes. Specifically, the study relied on the Chinese NGO (CNGO) dataset, which consists of 16 datasets containing various information related to NGOs, including financial reports and statistics on stakeholders and donations. After eliminating missing data, the study focused on data collected in 2016 for subsequent analysis.

The dataset used in the study comprises around 60 variables, including the number of donations (Donrev) and the position of the management board members in NGOs (Pos). This dataset is unique in that it also includes variables such as the characteristics of NGOs' stakeholders and long-term equity investments, which can offer new and valuable insights into the analysis. The initial dataset for this study comprised 6,589 samples and was obtained from the CNRDS, a high-quality research database that provides quantitative data. Specifically, the study utilized the CNGO dataset within the CNRDS, which consists of 16 datasets, including data from NGOs' financial reports, statistics on stakeholders and donations, and other related variables. After excluding samples with missing data, the data in 2016 was selected for further analysis. However, the raw dataset required cleaning due to various issues. A total of 402 samples were excessive, 1,776 contained missing data, 2,287 were problematic statistics, and the remaining samples lacked complete records. Consequently, 2,124 samples were deemed suitable for subsequent analysis. Adopting Momin's [62] classification, social NGOs advocate for human rights, poverty reduction, and other social welfare issues, while ENGOs focus on influencing policies related to environmental pollution. In the current research, a comprehensive analysis was carried out that included 146 ENGOs and 1,978 social NGOs.

## Data analysis

**Regression analysis.** In past studies, NGO professionalism has been related to a certain number of measurements. Again, according to García-Sánchez [63], efficiency was conceptualized as the rational use of resources to maximize benefits. Different from for-profit organizations, efficiency in the NGO sector is based on the improvement of social welfare [7]. In this regard, to measure efficiency, the amount of funding used to carry out projects was adopted as the indicator to measure social welfare. In this context, the dependent variable in the subsequent regression models was the program efficiency, which was measured by the proportion of funding used to carry out projects. In Model 1, the relationship between program efficiency and NGO professionalism is discussed, while models 2 address the linkage between program efficiency and political connections. Models 1 and 2 are presented as follows:

$$Efficiency = \alpha_1 + \beta_1 Feedback + \delta_1 Staff + \zeta_1 Income + \eta_1 Age + \theta_1 Scope + \iota_1 Start\_Fund \mp \gamma_1 Volunteer + \varepsilon_1 \quad (1)$$

**Table 2. Descriptive statistics.**

| | N | Min | Max | Mean | SD |
|---|---|---|---|---|---|
| Feedback | 2,095 | 0.00 | 5.00 | 1.32 | 1.79 |
| Staff | 2,095 | 1.00 | 35.00 | 3.81 | 3.24 |
| Volunteer | 2,095 | 0.00 | 5400.00 | 85.51 | 367.98 |
| Income | 2,088 | 0.73 | 8.51 | 6.97 | 0.71 |
| Pre_Gov | 2,095 | 0.00 | 5.00 | 0.08 | 0.35 |
| Current_Gov | 2,095 | 0.00 | 18.00 | 0.35 | 1.14 |
| Registration | 2,095 | 1.00 | 4.00 | 3.00 | 0.32 |
| Gov_Fund | 2,095 | 0.00 | 7.76 | 1.73 | 2.78 |
| Yr | 2,095 | 4.00 | 38.00 | 11.51 | 7.40 |
| Start_Fund | 2,095 | 5.60 | 8.20 | 6.57 | 0.33 |
| Scope | 2,095 | 1.00 | 2.00 | 1.05 | 0.21 |
| Efficiency | 2,088 | 0.39 | 7.08 | 0.88 | 0.16 |
| N | 2,088 | | | | |

$$Efficiency = \alpha_2 + \beta_2 Previous_{Gov} + \delta_2 Current\_Gov + \zeta_2 Registration + \eta_1 Age + \theta_1 Scope$$
$$+ \iota_1 Start\_Fund + \gamma_1 Gov\_Fund + \varepsilon_1, \quad (2)$$

In Models (1) and (2), $\alpha_i$ is the intercept, $\beta_j$ is the slope, and $\varepsilon_k \sim (0, \sigma_k^2)$ is the residual term, where $i,j,k = 0,1,2,\ldots,8$.

**Empirical results.** Table 2 presents the descriptive statistics of the variables used in the study before discussing the empirical results.

In particular, the level of governmental organization that granted the NGO license (Registeration), the number of staff members who have previously worked for the government (Pre_Gov), and the number of staff members who are currently working for the government (Current_Gov) were used to measure NGOs' political ties. The results suggest that within the NGOs sampled, an average of 0.08 and 0.35 of the staff have experience working either previously or currently for the government, respectively.

As previously mentioned, the professionalism of NGOs is assessed using various indicators such as the years since the organization's establishment (Yr), government evaluation of NGOs (*Feedback*), number of full-time staff (*Staff*), number of volunteers (*Volunteer*), and the regional scope in which NGOs operate (*Scope*). The descriptive statistics show that the average operation period for the sample NGOs is 11.51 years. Regarding government evaluation, most NGOs have a rating of 1.32A, with only a few certified with a 5A rating. The majority of NGOs have more than three full-time staff and 85 volunteers.

Besides, the descriptive statistics and correlations are presented in Table 3. A correlation analysis was conducted to examine the relationship between the dependent and independent variables, and the results were presented in Table 3. It can be seen that the number of full-time staff, amount of government fund and annual income are statistically and negatively correlated with NGO program efficiency.

In addition, the results of the multicollinearity test are illustrated in Table 4. When building a regression model, it is important to test for the multicollinearity of variables, as highlighted by Robinson and Schumacker [64]. Multicollinearity can lead to inflated variances of the variables, which can result in problematic regressions due to the limited new and independent information that the variables can provide Daoud [65]. To detect the degree of multicollinearity, the variance inflation factor (VIF) is often used, which measures the degree of correlation

**Table 3. Descriptive statistics and correlations.**

|  | Age | Feedback | Registration | Nproper | Nstaper | Scope | Volunteer | Staff | Income | Start_Fund | Gov_Fund | Efficiency |
|---|---|---|---|---|---|---|---|---|---|---|---|---|
| Age | 1.00 | | | | | | | | | | | |
| Feedback | 0.30** | 1.00 | | | | | | | | | | |
| Registration | 0.20** | 0.09** | 1.00 | | | | | | | | | |
| Current_Gov | 0.13** | 0.09** | 0.16** | 1.00 | | | | | | | | |
| Pre_Gov | 0.05* | -0.01 | -0.01 | 0.08** | 1.00 | | | | | | | |
| Scope | 0.14** | 0.04* | 0.69** | 0.21** | -0.02 | 1.00 | | | | | | |
| Volunteer | -0.033 | 0.05* | 0.02 | -0.02 | 0.01 | .04* | 1.00 | | | | | |
| Staff | 0.21** | 0.26** | 0.23** | 0.12** | -0.01 | 0.28** | 0.14** | 1.00 | | | | |
| Income | 0.25** | 0.36** | 0.21** | 0.11** | -0.01 | 0.19** | 0.08** | 0.35** | 1.00 | | | |
| Start_Fund | 0.23** | 0.11** | 0.33** | 0.13** | 0.06** | 0.41** | 0.03 | 0.19** | 0.33** | 1.00 | | |
| Gov_Fund | 0.2** | 0.17** | 0.02 | 0.12** | 0.12** | -0.01 | 0.01 | 0.12** | 0.22** | 0.15** | 1.00 | |
| Efficiency | 0.05* | 0.01 | 0.03 | -0.02 | -0.02 | -0.01 | -0.03 | -0.14** | -0.30** | -0.01 | -0.05* | 1.00 |

Note

**, and * denote significance at the 1% and 5% levels, respectively.

of an independent variable with the other independent variables [65–67]. Daoud [65] recommends using a VIF of 5 as the threshold for excessive multicollinearity. The findings indicate that all variables exhibited a VIF value below 3, thereby implying an absence of severe multicollinearity within the regression models.

Table 5 delineates the regression analysis outcomes for ENGOs, thereby elucidating a positive and statistically significant correlation between the number of full-time staff and the efficiency of the NGO. Concurrently, it also indicates a negative and significant association between their annual revenue and program efficiency. The inverse correlation between annual revenue, the quantity of full-time employees, and operational efficiency is also observable in the realm of social NGOs. It is of considerable importance to note that performance feedback plays a pivotal role in enhancing the efficiency of such social NGOs. Considering the findings from Model 2, the regression analysis unveiled an inverse relationship between the efficiency of both environmental and social NGOs and the magnitude of government funding they procure. In contrast to ENGOs, there is a significant and negative association between the efficiency of social NGOs and their level of registration. Within the context of social NGOs, those with superior registration levels (for instance, registered with national governmental agencies)

**Table 4. Multicollinearity test of independent variables.**

| Results | Model 1 | Model 2 |
|---|---|---|
| Feedback | 1.24 | |
| Staff | 1.26 | |
| Volunteer | 1.03 | |
| Income | 1.32 | |
| Current_Gov | | 1.07 |
| Pre_Gov | | 1.02 |
| Registration | | 1.97 |
| Gov_Fund | | 1.07 |
| Start_Fund | 1.13 | 1.10 |
| Age | 1.15 | 1.06 |
| Scope | 1.19 | 2.04 |

**Table 5. Regression statistics.**

| Panel A: Regression Statistics for Environmental NGOs | | |
|---|---|---|
| **Results** | **Model 1** | **Model 2** |
| Feedback | 0.01 (0.01) | |
| Staff | 0.01* (0.01) | |
| Volunteer | 0.01 (0.01) | |
| Income | -0.07*** (0.01) | |
| Current_Gov | | -0.01 (0.03) |
| Past_Gov | | -0.01 (0.01) |
| Registration | | 0.02 (0.06) |
| Gov_Fund | | -0.01 (0.04) |
| Age | 0.01 (0.01) | -0.01 (0.01) |
| Start_Fund | 0.01 (0.01) | -0.01 (0.01) |
| Scope | 0.01 (0.03) | -0.02 (0.72) |
| Adjusted $R^2$ | 0.40 | 0.01 |
| P-value | 0.01 | 0.99 |
| N | 117 | 117 |
| **Panel B: Regression Statistics for Social NGOs** | | |
| Results | **Model 1** | **Model 2** |
| Feedback | 0.01*** (0.01) | |
| Staff | -0.01*** (0.01) | |
| Volunteer | 0.01 (0.01) | |
| Income | -0.04*** (0.01) | |
| Previous_Gov | | -0.01 (0.01) |
| Current_Gov | | -0.01 (0.01) |
| Registration | | 0.02*** (0.01) |
| Gov_Fund | | -0.01*** (0.01) |
| Age | 0.01*** (0.01) | 0.01*** (0.01) |
| Start_Fund | 0.01*** (0.01) | 0.01 (0.01) |
| Scope | 0.04*** (0.01) | -0.03* (0.01) |
| Adjusted $R^2$ | 0.42 | 0.03 |
| P-value | 0.01 | 0.01 |

(*Continued*)

**Table 5.** (Continued)

| | | |
|---|---|---|
| N | 1,968 | 1,968 |

Note

**, and * denote significance at the 1% and 5% levels, respectively.

demonstrate greater efficiency when contrasted with entities possessing lower registration statuses (such as those registered with municipal governmental agencies).

In Table 6, the robustness test of the empirical results is presented. Based on Table 6, the results show that all outcomes are robust as one variable was replaced by another alternative variable in each model. Particularly, in models 1 and 2, the variable *Scope* which represents the regional scope for NGO's operation, was replaced by the variable *Staff* which indicates the number of full-time staff. As a result, it was found that the replacement of two variables in models 1 and 2 did not significantly alter the significance of the models. This indicates that the regression models are robust and the results are reliable.

## Discussion

Within business literature, research has been conducted comparing performance indicators across enterprises with varying ownership structures. For example, Lazzarini and Musacchio [68] analysis shows that state-owned enterprises (SOEs) perform similar to private firms, except when facing shocks like severe recessions that emphasize their social and political goals. Organizations with larger government stakes tend to emphasize governmental duty indicators over ethical and economic ones in their Performance Measurement Systems and compensation, while non-SOEs focus more on economic value indicators, neglecting societal measures [68, 69]. In summary, these studies highlight the varying impacts of different ownership structures on the key determinants of organizational performance. In this study, it was assumed that the factors influencing performance may differ among nonprofits, based on their mission focuses. Echoing Momin's [62] classification, social NGOs work towards human rights, poverty reduction, and other social welfare matters, while ENGOs concentrate on impacting policies pertaining to environmental pollution. Following these, the subsequent discussions are formulated.

In terms of the impact of NGO professionalism on efficiency, the initial regression model demonstrated a correlation between NGO professionalism and efficiency. The analysis suggests that positive feedback marginally enhances the efficiency of environmental and social NGOs. Theoretically, Baker et al. [70] conceptualized feedback as a continuous, interactive communication process conveying information about an individual's performance in work-related tasks. A multitude of studies have demonstrated that a discrepancy between an organization's actual and aspired performance—manifested in areas such as the intensity of research and development [71], the rate of new product introductions [72], and the extent of strategic changes [73]—can emerge when their performance surpasses targeted aspirations. Under the context of organizational studies, Ref and Shapira [74] illustrate that firms, when significantly under or over their aspiration level, show a marked change in behavior, reflected in an inverted U-shaped relationship with their likelihood to enter new markets. Besides, Kotiloglu et al. [75] posit that performance feedback influences investment and growth, strategic alterations, R&D intensity, and organizational risk-taking, but it does not catalyze product innovation. The results of our regression statistics are consistent with these notions.

In addition to performance feedback, we also assumed that NGOs with more full-time staff are more efficient. In NGOs, human resources primarily consist of full-time employees and

**Table 6. Robustness tests of regression statistics.**

| Panel A: Regression Statistics for Environmental NGOs | | |
| --- | --- | --- |
| **Results** | **Model 1** | **Model 2** |
| Feedback | 0.03<br>(0.09) | |
| Staff | 0.04*<br>(0.01) | 0.01<br>(0.01) |
| Volunteer | 0.01<br>(0.01) | |
| Income | -0.07***<br>(0.01) | |
| Current_Gov | | -0.02<br>(0.03) |
| Past_Gov | | -0.01<br>(0.01) |
| Registration | 0.01<br>(0.02) | 0.01<br>(0.03) |
| Gov_Fund | | -0.01<br>(0.01) |
| Age | 0.01<br>(0.01) | -0.01<br>(0.01) |
| Start_Fund | 0.01<br>(0.01) | -0.01<br>(0.01) |
| Scope | | |
| Adjusted R$^2$ | 0.41 | 0.05 |
| P-value | 0.01 | 0.68 |
| N | 117 | 117 |
| **Panel B: Regression Statistics for Social NGOs** | | |
| Results | **Model 1** | **Model 2** |
| Feedback | 0.01***<br>(0.01) | |
| Staff | -0.01***<br>(0.01) | -0.01***<br>(0.01) |
| Volunteer | 0.01<br>(0.01) | |
| Income | -0.04***<br>(0.01) | |
| Previous_Gov | | -0.01<br>(0.01) |
| Current_Gov | | -0.01<br>(0.01) |
| Registration | 0.04***<br>(0.01) | -0.03***<br>(0.01) |
| Gov_Fund | | -0.01*<br>(0.01) |
| Age | 0.01***<br>(0.01) | 0.01***<br>(0.01) |
| Start_Fund | 0.01***<br>(0.01) | 0.01<br>(0.01) |
| Scope | | |
| Adjusted R$^2$ | 0.22 | 0.16 |
| P-value | 0.01 | 0.01 |
| N | 1,968 | 1,968 |

Note:**, and * denote significance at the 1% and 5% levels, respectively.

volunteers. Yet, they are playing different roles in NGOs. According to Bowman [76], non-profit managers should not view volunteers as substitutes for paid staff. Ariza-Montes et al. [77] posit that paid staff, due to their professional training, hold positions of responsibility overseeing complex activities, while volunteers, determined by their availability and commitment, contribute to various tasks within a flexible schedule. Put simply, while volunteers provide valuable manpower in the nonprofit sector, their proficiency in handling everyday tasks can be limited. Therefore, it's arguable that full-time NGO staff are essential for boosting organizational performance. Nevertheless, the findings reveal a discrepancy in the correlation between the size of paid staff and efficiency across different NGO types. Specifically, while a positive correlation is observed between the size of paid staff and efficiency in ENGOs, a negative correlation emerges in the context of social NGOs. This divergence could potentially be attributed to the fact that an increased staff size may diminish staff participation in decision-making [78], a factor that may exert a negative influence in social-oriented nonprofits. Ecer et al. [7] delineate a positive correlation between a nonprofit organization's reliance on donations and its effectiveness. Yet, our findings indicate a significant negative correlation between the income of NGOs and their efficiency, with this association being notably stronger in ENGOs as compared to their social counterparts. One of the reasons might be that organizations may face pressure to allocate resources to building reserves for future repair and replacement needs, potentially hindering current performance [79].

Moreover, based on hypothesis 2, there exists a substantial body of literature in organizational studies that explores the impacts of political connections. Through empirical investigation, it was determined that political involvement can enable nonprofit foundations to secure additional government subsidies, attract more donations, and increase market revenue [57]. Similarly, Zhan and Tang [80] posit that ENGOs, when led by current government officials or legislative body members and maintaining robust guanxi, are likely to experience increased funding stability, a more sophisticated management system, and a higher propensity for policy advocacy. Based on the regression outputs, political connections are negatively related to NGO efficiency. Indeed, Coupet [81] suggested that government funding is a crucial input that can enhance organizational stability and aid in achieving significant social objectives. However, it can also increase costs to meet the requirements imposed by government agencies, potentially leading to a loss of uniqueness due to conforming to the guidelines provided by the government. In a similar vein, Jobome's [82] study found a positive correlation between government funding and governance requirements, as well as traditional charity structures, with efficiency. However, the adoption of business-type corporate governance codes does not show a similar relationship. In line with prior works, based on the regression statistics, it can be seen that social NGOs' efficiency is slightly and negatively influenced by the number of grants from government agencies. The reasons might be that obtaining government funding can be time-consuming [83] and such funding is often earmarked for particular uses [84]. Specifically, the level of government financial aid to NGOs is contingent upon the nature of their missions. According to Xie et al. [85], under the "big government, small society" program, Chinese ENGOs face an uneven distribution of government support, with sectors like nuclear security and the green economy finding it particularly challenging to secure funding. To put it differently, NGOs working on sensitive issues (e.g., social issues) that could be perceived as potential threats to the state may be more prone to restrictions imposed by the political system. Despite NGOs' positive impact, the government has curbed their growth [86]. They've developed legal tools to manage NGOs, viewing them as potential threats [87]. Thus, the inverse relationship between government funding and program efficiency is only seen in NGOs focused on social issues.

Additionally, Assenova and Sorenson [88] contend that legal registration is a crucial activity as it alters the legal status of a new business venture and bestows it with sociopolitical

legitimacy, which in turn, enhances the venture's access to various resources such as financial capital, human capital, raw materials, and customers. Relating this to our context, it can be seen from the results that social NGOs are more efficient when they register with senior government agencies. Such a linkage cannot be found in ENGOs. One of the reasons might be that, compared with ENGOs, the current registration system curtails the autonomy of NGOs, affecting their finances, personnel, and decision-making [89]. Chinese people may distrust NGOs because they represent special interests, which citizens regard as selfish [90]. However, the impact of this situation can be mitigated through the use of political connections. Indeed, in China, political trust significantly bolsters social trust [91].

NGOs primarily engage with state agencies to gain public trust, which Farid and Song [92] identified as essential for these groups to effectively operate their programs. In this sense, Johnson and Ni [60] found a modest yet positive correlation between the level of government funding received by an NGO (an indicator of legitimacy) and the amount of private donations it attracts. In terms of NGO project implication, Hildebrandt [43] suggests that NGOs leverage their political connections to expand their scope of activities beyond what is permitted by existing legal regulations. This study further reveals that the positive correlation between dual registration and NGO program efficiency is especially pronounced among social NGOs.

Moreover, in China, the non-profit sector is still relatively nascent, leading to an uncertain and fluid relationship between the state and the sector (Kang, 2017). This relationship is also dynamic and multifaceted [93], operating under frequently ambiguous political guidelines [89]. Indeed, China's NGO registration system created significant difficulties for NGOs' formation and operation. NGOs that managed to register under this dual management requirement were barred from creating local branches and had to undergo rigorous annual government reviews and reporting duties. Furthermore, the government employed subtle and indirect methods to regulate and control NGOs, such as offering funding or training, mandating the formation of communist party units within NGOs, and withholding essential resources [87]. Hence, registered NGOs have fostered self-censorship, focusing on obtaining government information and orderly participation [94]. They avoid actions that could be seen as challenging the government or threatening regime stability [95]. In addition, despite a corporatist regulatory system imposing strict registration and representational monopolies, Chinese ENGOs manage to function effectively due to self-regulation, international state pressure, and strong state-media relationships [96]. This could explain why a significant negative correlation among ENGOs cannot be observed in our context.

## Conclusion

This study aimed to investigate the determinants of NGO efficiency by analyzing the differences between environmental and social NGOs. Generally, it can be observed that there are differences in the influential factors of efficiency between environmental and social NGOs. Specifically, the regression analysis of ENGOs showed a positive and significant impact of the number of full-time staff on efficiency, while a negative and significant relationship was found between annual income and efficiency. Similar negative relationships between annual income, full-time staff number, and efficiency were also observed in social NGOs. Additionally, performance feedback was found to be a significant factor for social NGOs. The regression analysis of model 2 showed a negative correlation between the efficiency of both environmental and social NGOs and the amount of government funds they receive. However, in contrast to social NGOs, the efficiency of ENGOs was negatively and significantly associated with their registration level.

This study has made several significant contributions. Firstly, it has identified the influential factors of NGO efficiency and highlighted the differences between environmental and social NGOs, which contributed to the existing literature. Secondly, in terms of practical and social contributions, this study has provided valuable insights for NGO managers and policymakers in China to develop tailored measures to enhance efficiency for social and ENGOs (e.g., to improve program efficiency, compared with ENGOs, engagement of full-time staff in social NGOs). These findings have important implications for the development of the NGO sector in China and can guide policymakers in the field.

The present study is not without limitations. Firstly, the study is based on a secondary data-set obtained from the CNRDS database. As the data was collected in 2016, it may not reflect more recent developments or the current context, and thus could be considered somewhat outdated. For instance, the study does not consider the impact of the Covid-19 pandemic.

The second limitation is related to the complexities inherent to any attempt at quantifying political connections and their impacts. No universal consensus has yet been reached as to how such idiosyncratic social phenomena ought to be measured, and indeed, scholars such as Wang et al. [97] have all published studies that apply divergent indicators in pursuit of this elusive variable. The as-yet imperfect definition of political ties and their influences–a key variable in any analysis within this field–must be acknowledged. Future research in this vein must build upon previous models and seek to refine the definition of this crucial factor.

The third limitation concerns government feedback. As numerous researchers have recognised, feedback comes in many different forms, and its impacts can be just as diverse. For example, Swift and Peterson [98] propose that negative feedback can enhance motivation for conscientious and neurotic individuals during playful tasks, but conversely, it can hinder their motivation when tasks are frustrating. Thus, different types of feedback (e.g., positive and negative feedback) can lead to different consequences, which our model failed to account for. In this regard, future research should consider the impacts of different types of government feedback (e.g., positive and negative feedback) on NGO program efficiency.

## Author Contributions

**Data curation:** Sujie Peng.

**Formal analysis:** Sujie Peng.

**Funding acquisition:** Sujie Peng.

**Investigation:** Sujie Peng.

**Methodology:** Sujie Peng.

**Project administration:** Sujie Peng.

**Resources:** Sujie Peng.

**Software:** Sujie Peng.

**Validation:** Sujie Peng.

**Writing – original draft:** Sujie Peng.

**Writing – review & editing:** Sujie Peng.

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
