## [Decision Letter · Decision Letter 0]

29 Jan 2024

PONE-D-23-42144The Program Efficiency of Environmental and Social Non-governmental Organizations: A Comparative StudyPLOS ONE

Dear Dr. Peng,

Thank you for submitting your manuscript to PLOS ONE. After careful consideration, we feel that it has merit but does not fully meet PLOS ONE’s publication criteria as it currently stands. Therefore, we invite you to submit a revised version of the manuscript that addresses the points raised during the review process.

We look forward to receiving your revised manuscript.

Kind regards,

D. Daniel, Ph.D.

Academic Editor

PLOS ONE

“This work was supported by the UIC Start-up Research Fund [grant number UICR0700043-23].”

Reviewers' comments:

Reviewer's Responses to Questions

**Comments to the Author**

1. Is the manuscript technically sound, and do the data support the conclusions?

Reviewer #1: Partly

Reviewer #2: Partly

2. Has the statistical analysis been performed appropriately and rigorously? 

Reviewer #1: No

Reviewer #2: Yes

3. Have the authors made all data underlying the findings in their manuscript fully available?

Reviewer #1: Yes

Reviewer #2: No

4. Is the manuscript presented in an intelligible fashion and written in standard English?

Reviewer #1: Yes

Reviewer #2: Yes

5. Review Comments to the Author

Reviewer #1: In the hypothesis development part, it needs to be clarify why the research focuses on professionalism to measure efficiency and not the other factors such as the impacts of the organisations in delivering services to the community within limited resources. There are many previous research in other countries and China shows that professionalism, or in development word is called, NGO-ization, leads to inefficient and ineffective in downward accountability. Therefore, it would be good to explore about the contestation of professionalism in the literature review of hypothesis development before come with the results.

Reviewer #2: The Program Efficiency of Environmental and Social Non-Governmental Organizations: A Comparative Study

This study aims to investigate the factors influencing the program efficiency of NGOs in China and compare the differences between environmental and social NGOs

The author has not included findings from relevant and recent studies as this is dependent on the literature primarily before 2015. This is noted throughout the work. This requires updation.

Author fails to provide a demographics of the voluntary sector in China as many things needs to be understood in the context

Citation required Pg 11-‘However, various studies have discussed measurements of NGO efficiency’

Why the author wants to compare between environmental and social NGOs? Is there any specific reason which is comparable between? Also, in the results/discussion also this comparison/contrast is not outlined.

It will be good if the author could communicate the components with which he/she measured the efficiency? If there is a Govt rating, will that be the indicator of efficiency? And if yes, are those components taken into consideration for measurement?

Nproper and NStaper you mentioned the mean value is 0.01 which is not clear. What is that mean average? Is it the mean from the sum?

Unlike social NGOs, the efficiency of environmental NGOs is negatively and significantly associated with the registration level- What does that implies? Is the efficiency low for those organisations at low category of registration (Municipal)?

Likewise, it was also shown that managers who received higher and more consistent ratings from both themselves and others tended to have more positive outcomes (Ashford et al., 2018; Shull, 2010). The results of our regression statistics is consistent with these notions- This part is not clear. In the definition of variables, you have selected for this study, ‘feedback’ is defined as the variable which represent the NGOs evaluation from National body. Is this consistent with the rating of managers you have mentioned here?

6. PLOS authors have the option to publish the peer review history of their article (what does this mean?). If published, this will include your full peer review and any attached files.

Reviewer #1: **Yes: **Yusridar Mustafa

Reviewer #2: **Yes: **Kiran Thampi

---

## [Author Response · Author response to Decision Letter 0]

5 Mar 2024

Reviewer #1: 

1. In the hypothesis development part, it needs to be clarify why the research focuses on professionalism to measure efficiency and not the other factors such as the impacts of the organisations in delivering services to the community within limited resources. There are many previous research in other countries and China shows that professionalism, or in development word is called, NGO-ization, leads to inefficient and ineffective in downward accountability. Therefore, it would be good to explore about the contestation of professionalism in the literature review of hypothesis development before come with the results.

Reply: Thank you for your insightful comments. As detailed in Panel D of Table 1, the efficiency assessment is based on the percentage of funds that NGOs devote to the execution of projects central to their mission. The elements shown in Panels A and B represent potential factors impacting NGO efficiency. Additionally, I’ve introduced a new subsection on NGO professionalism within page 9 of the literature review, prior to the formulation of our hypothesis. Also, part of the hypothesis development has been updated.

Reviewer #2: 

2. This study aims to investigate the factors influencing the program efficiency of NGOs in China and compare the differences between environmental and social NGOs. The author has not included findings from relevant and recent studies as this is dependent on the literature primarily before 2015. This is noted throughout the work. This requires updation.

Reply: Thank you for your valuable suggestion. I have now updated the majority of the citations to reflect more recent sources (the sources in or after 2015). Nevertheless, a few studies, such as Kreutzer and Jäger (2011), Luo et al. (2012), and Hwang and Powell (2009), were not updated as these works established robust conceptual bases for specific concepts, were published in high-impact journals, and/or demonstrated substantial citation levels.

3. Author fails to provide a demographics of the voluntary sector in China as many things needs to be understood in the context

Reply: Thank you for your valuable input. I have added a new section to introduce the demographics of the voluntary (nonprofit) sector in China. This section can be found at the beginning of the literature review (page 5).

4. Citation required Pg 11-‘However, various studies have discussed measurements of NGO efficiency’

Reply: I am grateful for your constructive recommendation. In alignment with your advice, I have added the studies by Özbek (2015) and Polonsky et al. (2016) at the end of the sentence on page 2.

5. Why the author wants to compare between environmental and social NGOs? Is there any specific reason which is comparable between? Also, in the results/discussion also this comparison/contrast is not outlined.

Reply: Thanks for your valuable feedback. In response, I have elaborated on the rationale for comparing ENGOs and social NGOs at the conclusion of the introduction (page 4) and at the outset of the discussion (page 26). In particular, comparative research is crucial for understanding the macro-social variations in NGOs, revealing factors behind their global expansion and adaptation. Pioneering work by Lazzarini and Musacchio (2018) and Luong et al. (2019) shows that ownership structures affect organizational performance. However, following the Momin's (2013) classification, there is a research gap at the organizational level in non-western contexts, particularly between ENGOs (focused on sustainability and conservation) and social NGOs (addressing societal issues). Also, this research suggests that the determinants of performance (program efficiency) can differ among nonprofits, dependent on their mission focus. Thus, a comparative analysis of the efficiency of these two types of NGOs in China is necessary.

Additionally, we have enhanced the results and discussion sections by integrating a greater number of comparisons. For instance, in page 29, NGOs addressing social issues are often seen as potential threats by the state, leading to imposed restrictions. The government, despite recognizing the positive contributions of NGOs, has limited their expansion, using legal measures to control them. This has resulted in an inverse relationship between government funding and program efficiency, particularly in NGOs dealing with social issues.

6. It will be good if the author could communicate the components with which he/she measured the efficiency? If there is a Govt rating, will that be the indicator of efficiency? And if yes, are those components taken into consideration for measurement?

Reply: Thank you for your feedback. As shown in Panel D of Table 1, we measure efficiency by the proportion of funds that NGOs allocate towards the execution of mission-related projects. Furthermore, the performance feedback from the government and other components presented in Panels A and B are factors influencing NGO efficiency. They are not indicators of efficiency themselves.

7. Nproper and NStaper you mentioned the mean value is 0.01 which is not clear. What is that mean average? Is it the mean from the sum?

Reply: Thank you for your constructive suggestion. The incorrect average number concerning staff with work experience in governmental agencies has been updated to reflect the correct information as per Table 2. In particular, the findings propose that, within the scrutinized NGOs, a mean of 0.08 and 0.35 of the workforce have a background of either past or current employment with the government, respectively.

8. Unlike social NGOs, the efficiency of environmental NGOs is negatively and significantly associated with the registration level- What does that implies? Is the efficiency low for those organisations at low category of registration (Municipal)?

Reply: Thanks for your suggestion, in the main text, I have now added an explanation after this sentence. Specifically, it implies that: in terms of social NGOs, those possessing elevated registration levels (for instance, those registered at national governmental bodies) demonstrate enhanced efficiency relative to those with diminished registration levels (such as those registered at municipal governmental bodies).

9. Likewise, it was also shown that managers who received higher and more consistent ratings from both themselves and others tended to have more positive outcomes (Ashford et al., 2018; Shull, 2010). The results of our regression statistics is consistent with these notions- This part is not clear. In the definition of variables, you have selected for this study, ‘feedback’ is defined as the variable which represent the NGOs evaluation from National body. Is this consistent with the rating of managers you have mentioned here?

Reply: Thank you for your feedback. I have now incorporated references from organizational studies, specifically Kotiloglu et al. (2021) and Ref and Shapira (2017), to discuss how performance feedback can influence organizational behavior, such as improving NGO efficiency (page 27).

---

## [Decision Letter · Decision Letter 1]

2 Apr 2024

PONE-D-23-42144R1The program efficiency of environmental and social non-governmental organizations: A comparative studyPLOS ONE

Dear Dr. Peng,

Thank you for submitting your manuscript to PLOS ONE. After careful consideration, we feel that it has merit but does not fully meet PLOS ONE’s publication criteria as it currently stands. Therefore, we invite you to submit a revised version of the manuscript that addresses the points raised during the review process.

We look forward to receiving your revised manuscript.

Kind regards,

D. Daniel, Ph.D.

Academic Editor

PLOS ONE

Journal Requirements:

**Additional Editor Comments:**

Please see minor comments from the reviewer 2

Reviewers' comments:

Reviewer's Responses to Questions

**Comments to the Author**

1. If the authors have adequately addressed your comments raised in a previous round of review and you feel that this manuscript is now acceptable for publication, you may indicate that here to bypass the “Comments to the Author” section, enter your conflict of interest statement in the “Confidential to Editor” section, and submit your "Accept" recommendation.

Reviewer #1: All comments have been addressed

Reviewer #2: All comments have been addressed

2. Is the manuscript technically sound, and do the data support the conclusions?

Reviewer #1: Yes

Reviewer #2: Yes

3. Has the statistical analysis been performed appropriately and rigorously? 

Reviewer #1: Yes

Reviewer #2: Yes

4. Have the authors made all data underlying the findings in their manuscript fully available?

Reviewer #1: Yes

Reviewer #2: No

5. Is the manuscript presented in an intelligible fashion and written in standard English?

Reviewer #1: Yes

Reviewer #2: Yes

6. Review Comments to the Author

Reviewer #1: The elaboration on the connection between efficiency (in which measured through professionalism, political connection and funding) and how they affect NGOs operate in China to meet the need of governments and its people, makes the paper clearer.

Reviewer #2: Please order the reference in Alphabetical order. You need not to have sub headers under the 'discussion' section. Thank you for all the revisions recommended

7. PLOS authors have the option to publish the peer review history of their article (what does this mean?). If published, this will include your full peer review and any attached files.

Reviewer #1: **Yes: **Yusridar Mustafa

Reviewer #2: **Yes: **Dr Kiran Thampi

---

## [Author Response · Author response to Decision Letter 1]

2 Apr 2024

Reviewer #1: The elaboration on the connection between efficiency (in which measured through professionalism, political connection and funding) and how they affect NGOs operate in China to meet the need of governments and its people, makes the paper clearer.

Re: Thanks very much for your positive comment.

Reviewer #2: Please order the reference in Alphabetical order. You need not to have sub headers under the 'discussion' section. 

Re: Thanks for your comments, I would like to clarify that the references are not listed in alphabetical order due to the specific formatting requirements of the journal. Specifically, the journal requires that references are listed at the end of the manuscript and numbered in the order that they appear in the text. Therefore, the current ordering of references adheres to these guidelines. Moreover, I have substituted the subheadings with thematic sentences in the discussion portion of the paper.

---

## [Editor Report · Decision Letter 2]

15 Apr 2024

The program efficiency of environmental and social non-governmental organizations: A comparative study

PONE-D-23-42144R2

Dear Dr. Peng,

We’re pleased to inform you that your manuscript has been judged scientifically suitable for publication and will be formally accepted for publication once it meets all outstanding technical requirements.

Kind regards,

D. Daniel, Ph.D.

Academic Editor

PLOS ONE